# Effect of Soil Solution Properties and Cu^2+^ Co-Existence on the Adsorption of Sulfadiazine onto Paddy Soil

**DOI:** 10.3390/ijerph182413383

**Published:** 2021-12-19

**Authors:** Ziwen Xu, Shiquan Lv, Shuxiang Hu, Liang Chao, Fangxu Rong, Xin Wang, Mengyang Dong, Kai Liu, Mingyue Li, Aiju Liu

**Affiliations:** 1School of Agricultural Engineering and Food Science, Shandong University of Technology, Zibo 225049, China; 17864388719@163.com (Z.X.); lvshiquan24@163.com (S.L.); HSX1419791336@163.com (S.H.); rong3557622763@163.com (F.R.); wangxinsdlg@163.com (X.W.); Dongmengyer@163.com (M.D.); 2School of Resources and Environmental Engineering, Shandong University of Technology, Zibo 255049, China; chaoliang718@163.com (L.C.); kliu@sdut.edu.cn (K.L.); myli@sdut.edu.cn (M.L.)

**Keywords:** sulfadiazine, Cu^2+^ co-existence, paddy soils, adsorption, soil properties

## Abstract

Paddy soils are globally distributed and saturated with water long term, which is different from most terrestrial ecosystems. To better understand the environmental risks of antibiotics in paddy soils, this study chose sulfadiazine (SDZ) as a typical antibiotic. We investigated its adsorption behavior and the influence of soil solution properties, such as pH conditions, dissolved organic carbon (DOC), ionic concentrations (IC), and the co-existence of Cu^2+^. The results indicated that (1) changes in soil solution pH and IC lower the adsorption of SDZ in paddy soils. (2) Increase of DOC facilitated the adsorption of SDZ in paddy soils. (3) Cu^2+^ co-existence increased the adsorption of SDZ on organic components, but decreased the adsorption capacity of clay soil for SDZ. (4) Further FTIR and SEM analyses indicated that complexation may not be the only form of Cu^2+^ and SDZ co-adsorption in paddy soils. Based on the above results, it can be concluded that soil solution properties and co-existent cations determine the sorption behavior of SDZ in paddy soils.

## 1. Introduction

Overuse and the uncontrolled disposal of antibiotics have caused severe environmental problems, especially for the soil environment [1,2,3]. As one of the main crops in the world, rice is widely planted and feeds the majority of the world’s population; while over 92% of rice production, as a primary staple food, is in Asia [4]. In China, the planting area of rice is near 30 million ha, which occupies over 20% of the total farmlands [5]. Therefore, paddy soils have a high chance of exposure to antibiotic pollutants, such as sulfonamides [1]. However, most studies focused on the adsorption and transport behaviors of antibiotics in upland farmland, while few investigations were about the behavior of antibiotics in paddy soils.

The co-existence of heavy metals and antibiotic pollution in the soil environment is receiving more and more attention. With their wide antibacterial spectrum, excellent curative effect, and low cost, sulfonamide antibiotics (SAs) are widely used in disease treatment and prevention for humans and animals [6]. Sulfadiazine (SDZ), one of the most commonly used SA chemicals, has been widely detected in various environmental mediums, especially in soils [1,7,8]. Previous studies proved that Cu ions could coordinate with SDZ [9,10,11], altering their molecular speciation and environmental behavior [11,12]. For example, it was reported that Cu^2+^ co-addition could improve the adsorption of antibiotics to organic matters [13] and soils [14]. On the other hand, there was competitive adsorption between co-existent metal ions and the ionizable antibiotics, which inhibited the adsorption ability of antibiotics on soil components [15,16] and increase their transportation risk in the environment [17]. However, limited investigations have been conducted on the effect of Cu^2+^ on SA sorption in paddy soils.

To obtain more knowledge about the environmental risk of SAs in paddy soils, we chose Sulfadiazine (SDZ) as a representative antibiotic. The batch sorption experiments were conducted in soil suspensions with different pH, ion concentrations (IC), and dissolved organic carbon (DOC), as well as co-existent Cu^2+^. The aim of this study was: (1) to explore the sorption kinetics and isotherm of SDZ in paddy soils under different soil solution conditions; (2) to characterize the sorption behavior and mechanisms of SDZ on various soil components; and (3) to investigate the effects of co-existent Cu^2+^ on the sorption of SAs in paddy soils.

## 2. Materials and Method

### 2.1. Paddy Soil Sampling and Characterization

Five soil samples were collected from the rice production area located in Huang gang, Hubei, of China. About 20 kg of topsoil (0–10 cm) was collected from sample sites occupying an area of approximately 500 m^2^ and transported to the lab in a cabinet with an ice pack. After thorough mixing, soils were air-dried, ground, and sieved through a 2-mm sieve for the following soil property analysis and sorption batch experiments. The soil texture was silty loam with 2.55% clay, 89.67% silt, and 7.79% sand. Soil pH, organic matter (OM), cation exchange capacity (CEC), and Zeta potential are listed in Table 1.

### 2.2. Chemicals and Reagents

In this study, sulfadiazine (4-amino-N-5-methylisoxazol-3-yl)-benzene sulfonamide, (CAS: 723-46-6, purity >98%, MW 250 g mol^−1^) was purchased from Sigma-ALDRICH (Shanghai, China), whose lgK_ow_ and pKa values were −0.09, 2.00 (pK_a1_) and 6.50 (pK_a2_), respectively [18]. SDZ was first dissolved in methanol (grade: HPLC, Sigma, Shanghai, China) and diluted to 100 mg kg^−1^, with ultra-pure water used for a stock solution. Cu^2+^ stock solution (10 g·L^−1^) was prepared from CuCl_2_·2H_2_O (min. purity 99.99%, Sinopharm, Shanghai, China) in ultra-pure water. All the other chemicals, including Hydrochloric acid (HCl) (Sinopharm, Shanghai, China), sodium hydroxide (NaOH) (Sinopharm, Shanghai, China), calcium chloride (CaCl_2_) (Sinopharm, Shanghai, China), and ammonium dihydrogen phosphate (ADP) (Sinopharm, Shanghai, China), etc., used in this study were analytical grade, apart from acetonitrile (ACN) (Sigma, Shanghai, China), which was HPLC grade and used for the HPLC analysis of SDZ.

### 2.3. Preparation of Soil Organic Particles and Soil Clay

In this study, the soil organic particle fraction was obtained with the wet sieving method suggested by Elliott [19]. Briefly, put 500 g of air-dried soil on a sieve of 2 mm, and separate aggregates by shaking the sieve up and down for 2 min with 50 repetitions, after being submerged in water for 5 min. Aggregates passed through a 0.25-mm sieve were collected and designated as a soil organic particle fraction. The physical and chemical properties of the prepared organic particles are listed in Table 1.

The soil clay fraction was extracted with 0.1 M Na_4_P_2_O_7_ (pH = 7) and 0.1 M H_2_O_2_ according to [20]. After being treated with Na_4_P_2_O_7_ (Sigma, Shanghai, China) and 0.1 M H_2_O_2_ (Sinopharm, Shanghai, China), the sample mixture was centrifuged and separated into three distinct layers. After removing the top layer using suction, the residual material was separated into two fractions by repeat centrifuging. After removing the top liquid layer, recover the clay mineral fraction of 1 M NaCl (Sinopharm, Shanghai, China), after repeat centrifuging. The physical and chemical properties of the prepared clay are listed in Table 1.

### 2.4. Batch Sorption Experiment

Sorption experiments were conducted in 100-mL plastic centrifuge tubes with Teflon lids, according to the OECD 106 method and Jiang [21]. Briefly, weigh 1.000 g dry paddy soil into 25 mL of 0.01 M CaCl_2_ (Sinopharm, Shanghai, China) containing a specific concentration of SDZ. All sorption experiments were carried out with an oscillator ( Changzhou Guohua Electric Appliance Co., Ltd., Changzhou, China) at 150 rpm in darkness. The sample tubes containing an initial concentration of 8 μmol L^−1^ SDZ were shaken from 0 h to 72 h for the kinetic sorption experiment, and those containing a series of initial concentrations of 4 μmol L^−1^, 8 μmol L^−1^, 12 μmol L^−1^, and 16 μmol L^−1^ were shaken for 72 h for the isotherm experiment. At the end of the sorption experiments, 2 mL supernatant was collected from each tube and centrifuged at 10000 r for 1 min. Then, the supernatant was filtered through a 0.45 μm Whatman filter (Whatman^TM^, Germany) for the subsequent quantification with HLPC. Each experiment was conducted in triplicates, and a blank treatment of SDZ solution without soil was used as a control, which was used to evaluate the loss of SDZ caused by sorption onto the tube walls and its degradation.

*Environmental factors:* The pH of soil solution was adjusted to 5.0, 7.0, and 9.0, respectively, with 1 M HCl (Sinopharm, Shanghai, China) and NaOH (Sinopharm, Shanghai, China) to analyze the influence of the medium pH. Fulvic acid (FA, ≥90%, CAS: 1415-93-6, Aladdin, Shanghai, China) was added at rates of 1 g L^−1^, 3 g L^−1^, and 5 g L^−1^ to the soil solution to analyze the effect of soil DOC on the adsorption of SDZ on soils. At the same time, the effect of solution ionic concentration was evaluated with Ca^2+^ ions at the concentrations of 0.05 M and 0.1 M. The sorption characteristics of SDZ on various soil fractions was tested with the same procedure as the above soils, and the solid phase was replaced with the soil organic particles and clays prepared in 1.3.

*Effect of co-existent Cu^2+^*: For the sorption experiments of SDZ with Cu^2+^ co-existence, the added concentrations of Cu^2+^ were 200 and 500 mg L^−1^, respectively. The others were the same as the soil sorption experiment.

### 2.5. Analysis Method

The pH, CEC, and OM of the soil samples were analyzed according to the description of Lu (1999) [22]. The zeta potentials of the crude soil, organic particle fraction, and soil clay fraction were measured with a zeta potential analyzer (JS94H, Shanghai Zhongchen Digital Technic Apparatus Co., Ltd., Shanghai, China) after being dispersed in a solution of 0.01 M NaCl (Sinopharm, Shanghai, China). 

*HLPC analysis of SDZ:* SDZ concentration in the filtrate was quantified at 270 nm by a HPLC/UV (Agilent/Bruker HP1100/Esqure2000, Agilent/Bruker Co., Ltd., Walter cloth, Germany) using a C18 column. The mobile phase was ADP (0.01 M, pH = 2.8) (Sinopharm, Shanghai, China): CAN (Sinopharm, Shanghai, China ) (80:20, *v*:*v*) at a flow rate of 0.5 mL min^−1^. The injection volume was 10 μL. 

The concentration of SDZ was increased to 40 μmol L^−1^ in the soil solutions and they were then centrifuged after the adsorption reached equilibrium. The centrifuged samples were washed three times with ultra-water, dried, and powdered with a mortar and pestle for the subsequent scanning with FTIR spectrum and SEM visualization. The FTIR scanning was conducted with a FTIR Microscope-Spectrometer (Nicolet5700, Thermo, Shanghai, China) in the range 4000–400 cm^−1^. The microstructure observation of soil samples was conducted with a SEM (Quanta250, FEI, Shanghai, China) at magnifications of ×20,000.

### 2.6. Data Calculation

The amount of adsorbed SDZ was calculated using the following equation:(1)$$Y=\frac{V(C_{0}-C_{e})}{M}$$
where *Y* is the absorbed amount of SDZ (μmol kg^−1^); *V* (L) is the volume of soil solution; *C*_0_ and *C_e_* (μmol L^−1^) represent the concentration of SDZ at the beginning and end of the equilibrium sorption experiments in solution, respectively; and *M* (kg) is the dry soil weight added to the background solution.

The sorption kinetics of SDZ in paddy soils in various soil solutions were fitted with the following kinetic model equations:(2)$$q_{t}=q_{e}\left(1-e^{-k_{1}t}\right)$$
(3)$$q_{t}=\frac{k_{2}q_{e}^{2}t}{1+k_{2}q_{e}t}$$

The sorption isotherm of SDZ in paddy soils in various soil solutions were fitted with the Linear model equation (4) and Freundlich model equation (5):(4)$$q_{e}=K_{d}C_{eq}$$
(5)$$q_{e}=K_{F}C_{E}^{1/n}$$

For Equations (2)–(5), *q_t_* (μmol kg^−1^) is the amount of SDZ sorption to paddy soil at time t (h); *q_e_* (μmol kg^−1^) is the amount of SDZ sorption to paddy soil when sorption reaches the equilibrium; and *k_1_*/*k*_2_ is the constant of the kinetics sorption velocity. *C_e_* (μmol L^−1^) is the concentration of SDZ in the soil supernatant when sorption reaches equilibrium; *K_d_* (L kg^−1^) is the coefficient of SDZ distribution between the liquid and solid phases in the equilibrium system; *K_F_* is the Freundlich sorption coefficient; and *n* is the nonlinearity factor.

## 3. Results

### 3.1. Effect of Soil Solution Properties on the Sorption of SDZ

Batch sorption experiments were conducted to analyze the sorption characteristics of SDZ on paddy soils under different experimental systems. The results are shown in Figure 1.

As plotted in Figure 1b, the adsorption capacity of SDZ in crude soils was decreased, with the pH changing to 5.0, 7.0, and 9.0, and *q_e_* was over 60 μmol kg^−1^ in crude soil but lower than 50 μmol kg^−1^ with the pH change (Table 2). As for the ion strength, the adsorption capacity of SDZ on paddy soils was significantly reduced, to about 20 μmol kg^−1^, when the CaCl_2_ concentration increased to 0.05 M and 0.1 M CaCl_2_ (Figure 1d). The variation in the fitting parameters of the adsorption isotherms (Table 3) indicated that the soil solution pH and ion strength could change the sorption affinity of SDZ in paddy soil. However, there was a significant increase in SDZ adsorption capacity with increasing soil solution DOM (Figure 1f and Table 2). In addition, the values of K_1_ and K_2_ were enhanced with the increasing of pH and DOM (Table 2), which indicated that the increase of pH and DOM accelerated the reaction process and increased the velocity of SDZ adsorption on paddy soil (Figure 1a,c,e). Whatever the changes in soil solution properties in the present study, the adsorptions of SDZ had a good fit with the pseudo-first, pseudo-second-order, and the Freundlich equation, as the *R* parameter of each selected fitting model was always over 0.99 at a level of *p* < 0.05 (list in Table 2 and Table 3). This might suggest that there was more than one dominant mechanism responsible for the adsorption of SDZ.

### 3.2. Effect of Cu^2+^ Co-Existing on the Sorption of SDZ

The adsorption kinetics and isotherms of SDZ were compared for the crude soil, organic particles, and clay with the presence or absence of Cu^2+^ (Figure 2). Whether Cu^2+^ was present or not, the adsorption of SDZ on the three soil fractions could reach equilibrium within about 48 h (Figure 2a,c,e). However, with the co-existence of Cu^2+^, the adsorption capacity of SDZ on the crude soil and the organic fraction was significantly increased, which did not happen with the clay (Figure 2f). The values of *K_F_* and *n* were also enhanced with Cu^2+^ co-addition for the crude soils and organic particles. Morover, with the co-existence of Cu^2+^, the adsorption characteristics of SDZ on each soil fraction still fit well with the selected kinetic equations and isotherm equations, as most of the coefficients of R were >0.99 (Table 4 and Table 5), although the adsorption of SDZ was decreased for clay soils (Table 5). Moreover, the subsequent linear model fitting also found that the adsorption affinity (*k_d_*, Table 5) of clay to SDZ was much lower than the crude soil and organic particles, and the presence of Cu further decreased this adsorption affinity. Nevertheless, this was increased for the crude soil and organic particles.

### 3.3. Co-Adsorption Mechanism of Cu and SDZ on Different Soil Components

Figure 3 shows the FTIR spectra of soil OC and clay particles with/without SDZ/Cu adsorption. Compared to the treatment without SDZ, attenuation was observed at wavenumbers from 1050 to 950 cm^−1^ (C-O-C) on the OC particles after SDZ adsorption. Compared to the single adsorption of SDZ, a stretching vibration in the range 1700–1600 cm^−1^ (C=C or -NH) in OC components became stronger with Cu co-addition. These results suggest that the -NH, C-O-C, or C=C groups play an important role in the SDZ adsorption to soil particles [23]. For the clays, a weak peak, ranging from 3000 cm^−1^ to 2900 cm^−1^ (-CH), occurred after SDZ adsorption, and this signal became stronger in clay particles with the co-presence of Cu. Moreover, an attenuation of the wavenumbers from 1700 to 1600 cm^−1^ and an obvious -OH stretching vibration (3678.0 cm^−1^) were observed for the Cu-SDZ co-adsorption, compared to the single adsorption of SDZ. This might suggest that hydrogen bonding plays an essential role in the co-adsorption of Cu and SDZ on soil clays.

The surface morphologies of the soil OC and clay particles were scanned using SEM (Figure 4). The SEM images show that soil OC particles surface was rougher than the clay soils, with an irregular and bulky shape. However, there was no obvious change in the surface morphology of the OC and clay particles with the sorption of Cu and SDZ. According to the mapping results, the Cu was very hard to detect in OC and clay without Cu^2+^ treatment, but it covered the surface of the samples treated with Cu^2+^, with the signal becoming stronger for the combined organic soil treated with SDZ. This indicated that Cu^2+^ adsorbs easily on organic soil and clays, and SDZ co-addition increases the sorption of Cu on organic soil. From the above results of the FTIR analysis and the batch adsorption experiments, it was indicated that co-adsorption by the complexation of Cu^2+^-SDZ is one method of Cu^2+^ and SDZ adsorption to soil particles. While the co-adsorption of Cu^2+^ and SDZ might mainly be conducted by their independent adsorption to different sites on the soil particles.

## 4. Discussion

### 4.1. Effects of Soil Solution Properties on the Sorption of SDZ in Paddy Soils

In addition to the basic properties of antibiotics, soil physical and chemical properties also play an essential role in the sorption of antibiotics to soil constituents [24]. In this study, the adsorption kinetics and isotherms processes of SDZ were all well fitted (R > 0.99) by the first/second-order equations and the Freundlich equation, whatever the variation of the soil properties. This indicated that the variation of soil solution properties did not change the process and mechanism of SDZ adsorption on paddy soils, which might be ascribed to the hydrophobic distribution that contributed to the sorption of SDZ on paddy soil [21,25], in view of the chemical character of SDZ, which was amphoteric and weak acid polar, with little water-solubility. It was also proven that the adsorption process of SDZ in soils was driven by weak hydrophobic forces, as in neutral or anionic specie of natural soil. The present study also indicated further that the SDZ chemicals were easily adsorbed to the soil organic matter, as the SDZ adsorption increased with soil organic matter addition (Figure 1f), and there was more obvious signal noise for soil OC constituents than the clay after being treated with SDZ (Figure 3). These results indirectly suggests that hydrophobic distribution has a beneficial effect on the adsorption of SDZ in paddy soils.

The pH determined whether SDZ existed as cations, zwitterions, or anions in the soils [26]. Hence, SDZ exhibited pH-dependent adsorption on soil constituents [27,28]. The variation of soil solution pH significantly decreased the adsorption capacity of SDZ on paddy soils, based on the adsorption coefficients of SDZ (Table 3); which is similar to the previous results, that the increase of pH from 4 to 8 significantly decreased adsorption coefficients of SAs from 30 to 1 [29]. This might be explained by the fact that the disturbance of the acid–base balance of crude soils would increase the electrostatic repulsion between anionic SDZ^-^ and the negatively charged soil surface [25,30], and by the lower lipophilic interactions between the uncharged chemical molecules and soil particles [31]. As for the ion strength factor, a lower adsorption potential of SDZ was found with increasing Ca^2+^ concentration (Figure 1), which was dissimilar to the previous reports that multivalence ions can improve antibiotic adsorption by covalent bonding action [32]. This inconsistent result might indicate that a high ion strength would weaken the function of the ion-bridge for SDZ adsorption on paddy soils, as it has a higher OM content (Table 1).

### 4.2. Effect of Cu^2+^ Co-Existing on the Sorption of SDZ

With increased detection of heavy metals and antibiotics in soils, more attention has been paid to the process of antibiotics related to the co-existent metal ions, aside from the various soil properties [33,34,35]. It is widely thought that heavy metal cations may influence the mobility of antibiotics in soils, through ion exchange or complexation. In the present study, the adsorption isotherms of SDZ were well described by the Freundlich model (R = 0.9445–1.0000) in a binary system of Cu–SDZ. Moreover, both the adsorption capacity and affinity of SDZ were augmented, as Cu^2+^ was increased, based on the changing trend of *k_F_* and *n* values (Table 5). These results indicated that the presence of co-existent Cu^2+^ did not change the mechanism of SDZ adsorption to soil particles but could alter the adsorption capacity of SDZ to soil particles with different compositions.

The further FTIR and SEM analyses also detected the co-adsorption of Cu and SDZ on soil constituents (Figure 3 and Figure 4). It was suspected that a Cu^2+^ bridge between soil particles and SDZ [11,36] or the formation of SDZ–Cu complexes [37,38] occurred for the co-adsorption of Cu and SDZ on paddy soils. However, this might be mainly carried out by their independent adsorption to different sites of the soil particles, as the weak signal changes of the FTIR spectrum were found primarily in soil OC constituents after co-treating with Cu and SDZ (Figure 3). It was reported that the presence of heavy metal ions could result in changes in soil properties [39,40,41], which might shape the sorption of antibiotics on soils. Therefore, the decrease of SDZ adsorption on clay in the present study might be attributed to the precipitation of soil mineral colloid induced by the presence of Cu^2+^ [39], which decreased the size or blocked the pores of the soil clay particles [42].

## 5. Conclusions

The change of soil solution pH and ion concentration lowered the SDZ adsorption on paddy soil, but it was increased with the addition of organic matter. Whatever the changes of soil properties, the sorption kinetics of SDZ on paddy soils could be well described by the pseudo-first and second-order equations, and the Freundlich equation could fit the sorption isotherms well. Cu^2+^ coexistence in the soil increased the adsorption of SDZ on crude soil and its organic components, but decreased the adsorption capacity of clay soil for SDZ. Based on a further analysis of FTIR and SEM, it could be concluded that the co-adsorption of Cu^2+^ and SDZ on soil constituents might be conducted by the complexation of Cu^2+^ and SDZ, in addition to their independent adsorption to different active sites of the soil particles.

## Figures and Tables

**Figure 1 ijerph-18-13383-f001:**
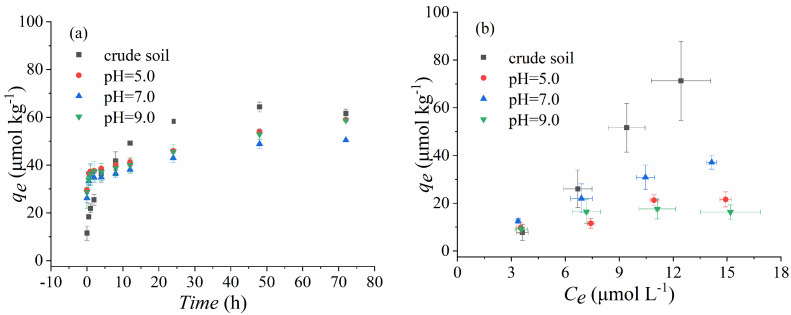
The sorption kinetics (**a**,**c**,**e**), and isotherm (**b**,**d**,**f**) of SDZ in soil solution with different pH, IC, and DOM conditions.

**Figure 2 ijerph-18-13383-f002:**
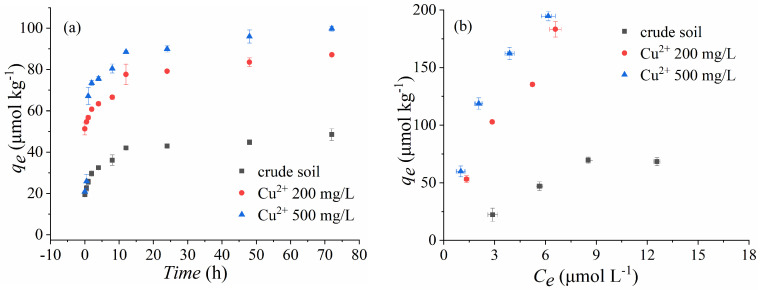
The sorption kinetics (**a**,**c**,**e**) and isotherms(**b**,**d**,**f**) of SDZ in the crude soil, organic particles, and clay with Cu^2+^ co-existing in the soil solution.

**Figure 3 ijerph-18-13383-f003:**
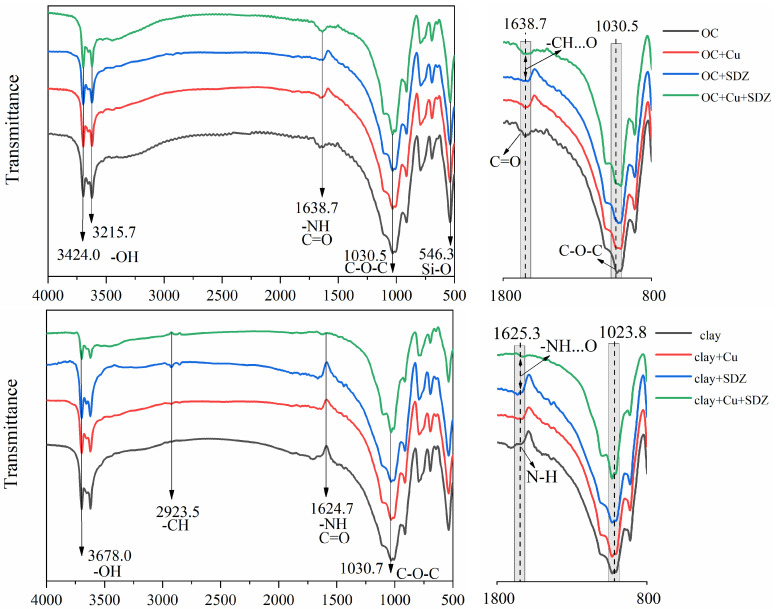
The FTIR spectra of OC and clay before and after the reaction with SDZ and Cu^2+^.

**Figure 4 ijerph-18-13383-f004:**
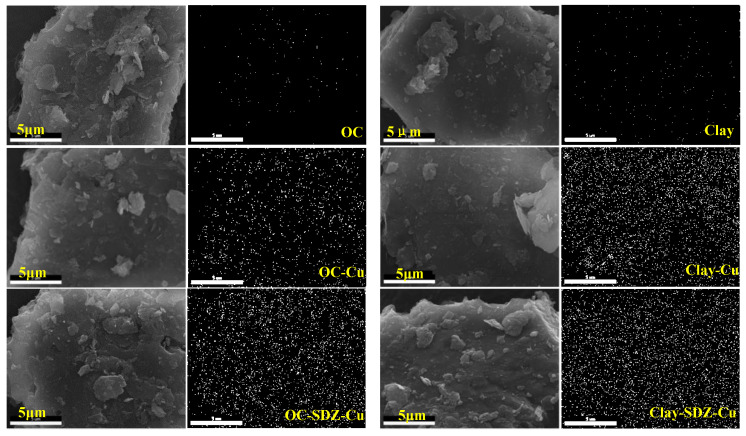
SEM images of OC and clay before and after reaction in Cu^2+^ and SDZ, elemental mappings showing the distributions of Cu.

**Table 1 ijerph-18-13383-t001:** Physical and chemical properties of paddy soils in this study.

Soil Components	pH	OM ^1^ (mg g^−1^)	CEC ^2^ (cmol kg^−1^)	Zeta-Potential (ζ, mV)
Crude soil	6.8	23.68 ± 2.46	5.84 ± 0.11	−33.79 ± 2.06
Organic particle	7.3	50.55 ± 3.12	8.15 ± 0.19	−59.99 ± 5.25
Soil Clay	5.3	--	3.84 ± 0.04	−64.55 ± 3.35

^1^, OM: organic matter; ^2^, CEC: cation exchange capacity.

**Table 2 ijerph-18-13383-t002:** Fitting results of kinetics data to pseudo-first and pseudo-second-order equations for SDZ adsorption in various soil solutions (mean values ± standard error).

Treatments	The Pseudo-First-Order			The Pseudo-Second-Order		
	*q_e_* (μmol kg^−1^)	*k* _1_	*R*	*q_e_* (μmol kg^−1^)	*k* _2_	*R*
Crude soil	65.84 ± 0.36	1.40 ± 0.59	0.9980	69.48 ± 0.76	0.02 ± 0.00	0.9901
pH 5.0	60.16 ± 0.37	0.19 ± 0.02	0.9990	65.72 ± 0.32	0.02 ± 0.00	0.9996
pH 7.0	41.56 ± 0.30	1.69 ± 0.69	0.9986	43.96 ± 0.33	0.20 ± 0.07	0.9989
pH 9.0	45.72 ± 0.40	1.04 ± 0.37	0.9977	48.40 ± 0.45	0.13 ± 0.05	0.9983
FA 1 mg L^−1^	52.08 ± 0.33	0.33 ± 0.15	0.9986	74.24 ± 0.39	0.04 ± 0.01	0.9990
FA 3 mg L^−1^	69.24 ± 0.44	0.45 ± 0.08	0.9979	74.24 ± 0.39	0.04 ± 0.01	0.9990
FA 5 mg L^−1^	71.12 ± 0.29	0.29 ± 0.13	0.9989	74.52 ± 0.25	0.07 ± 0.01	0.9995
0.05 M CaCl_2_	22.88 ± 0.27	0.06 ± 0.01	0.9997	28.28 ± 0.49	0.49 ± 0.01	0.9998
0.1 M CaCl_2_	19.68 ± 0.20	0.16 ± 0.02	0.9999	22.00 ± 0.19	0.19 ± 0.01	0.9999

**Table 3 ijerph-18-13383-t003:** Fitting results of the Linear and Freundlich models for adsorption curves of SDZ in various soil solutions (mean values ± standard error).

Treatments	Linear		Freundlich		
	*k_d_* (L kg^−1^)	*R*	*K_F_* (μmol^1 − N^ L^N^ kg^−1^)	*n*	*R*
Crude soil	2.17 ± 0.47	0.9537	18.56 ± 1.95	1.48 ± 0.09	0.9990
pH 5.0	1.65 ± 0.11	0.9743	12.57 ± 0.75	1.22 ± 0.26	0.9755
pH 7.0	1.82 ± 0.15	0.9841	17.47 ± 0.86	1.32 ± 0.46	0.9952
pH 9.0	1.42 ± 0.17	0.9251	11.85 ± 0.66	1.32 ± 0.42	0.9578
FA 1 mg L^−1^	2.07 ± 0.21	0.9436	24.46 ± 0.68	1.89 ± 0.20	0.9644
FA 3 mg L^−1^	3.70 ± 0.33	0.9540	25.56 ± 0.73	1.89 ± 0.86	0.9925
FA 5 mg L^−1^	4.22 ± 0.36	0.9589	28.45 ± 0.96	1.96 ± 0.30	0.9950
0.05M CaCl_2_	1.48 ± 0.13	0.9538	10.86 ± 0.35	1.90 ± 0.40	0.9772
0.1 M CaCl_2_	4.22 ± 0.36	0.9589	10.90 ± 0.54	1.27 ± 0.40	0.9454

**Table 4 ijerph-18-13383-t004:** Fitting results of kinetics data to pseudo-first and pseudo-second-order equations for SDZ adsorption on soil composition.

Treatments	The Pseudo-First-Order			The Pseudo-Second-Order		
	*qe* (μmol L^−1^)	*k* _1_	*R*	*qe* (μmol L^−1^)	*k* _2_	*R*
Crude soil	42.76 ± 0.26	0.54 ± 0.09	0.9992	45.64 ± 0.23	0.07 ± 0.01	0.9996
Cu^2+^ 200 mg L^−1^	77.92 ± 0.42	0.76 ± 0.18	0.9977	82.40 ± 0.40	0.06 ± 0.01	0.9988
Cu^2+^ 500 mg L^−1^	91.16 ± 0.37	0.58 ± 0.10	0.9984	93.92 ± 0.71	0.07 ± 0.01	0.9178
Organic particles	70.64 ± 0.75	0.09 ± 0.01	0.9974	81.64 ± 0.97	0.01 ± 0.00	0.9982
Cu^2+^ 200 mg L^−1^	79.52 ± 0.51	0.34 ± 0.06	0.9975	104.56 ± 0.49	0.03 ± 0.01	0.9987
Cu^2+^ 500 mg L^−1^	81.60 ± 0.40	0.39 ± 0.07	0.9984	106.08 ± 0.36	0.04 ± 0.01	0.9992
Clay	19.40 ± 0.11	0.25 ± 0.04	0.9999	20.36 ± 0.11	0.14 ± 0.03	1.0000
Cu^2+^ 200 mg L^−1^	15.24 ± 0.26	0.09 ± 0.01	0.9997	18.96 ± 0.36	0.02 ± 0.00	0.9997
Cu^2+^ 500 mg L^−1^	13.76 ± 0.14	0.66 ± 0.14	0.9998	15.08 ± 0.13	0.22 ± 0.05	0.9999

**Table 5 ijerph-18-13383-t005:** Fitting results of the Linear and Freundlich models for adsorption curves of SDZ on soil composition (mean values ± standard error).

Treatments	Linear		Freundlich		
	*k_d_* (L kg^−1^)	*R*	*K_F_* (μmol^1^ L^N^ kg^−1^)	*n*	*R*
Crude soil	6.57 ± 0.38	0.9801	18.65 ± 1.56	1.45 ± 0.29	0.9992
Cu^2+^ 200 mg L^−1^	28.62 ± 1.09	0.9914	35.76 ± 2.34	1.20 ± 0.12	1.0000
Cu^2+^ 500 mg L^−1^	36.62 ± 2.62	0.9706	40.68 ± 2.66	1.87 ± 0.25	1.0000
Organic particles	1.69 ± 0.20	0.9266	15.43 ± 1.35	1.46 ± 0.06	0.9938
Cu^2+^ 200 mg L^−1^	9.05 ± 0.94	0.9408	30.21 ± 1.98	1.81 ± 0.32	0.9987
Cu^2+^ 500 mg L^−1^	11.55 ± 0.85	0.9692	32.53 ± 2.01	1.13 ± 0.30	0.9996
Clay	1.87 ± 0.13	0.9725	13.69 ± 0.96	1.63 ± 0.08	0.9940
Cu^2+^ 200 mg L^−1^	1.66 ± 0.12	0.9715	12.56 ± 0.58	1.54 ± 0.04	0.9979
Cu^2+^ 500 mg L^−1^	1.00 ± 0.15	0.8920	8.57 ± 0.35	1.50 ± 0.14	0.9445

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
