# Peer review of "Effect of Soil Solution Properties and Cu2+ Co-Existence on the Adsorption of Sulfadiazine onto Paddy Soil"

_ijerph, 2021, doi:10.3390/ijerph182413383_

Round 1

Reviewer 1 Report

Dear Authors, please take into account the following comments:

  • The title of the paper should be modified, avoiding the term "affect" as it is vague
  • Further explanations should be included in the text about the reasons why Sulfadiazine was chosen as a SA representative. 
  • Relocate Figure 1 so it is shown as soon as it is referenced in the text (section 2.2) and not later
  • There are many english typos and grammar mistakes throughout the manuscript 

Reviewer 2 Report

The paper under the title "Soil solution properties and Cu2+ co-existence affect adsorption of sulfadiazine on paddy soil" describes the behaviour of the widely-used antibiotic in the soil environment, particularly in paddy soil. The Authors did a great job performing the wide scope of sorption studies, supported by the spectroscopic analyses. The results are interesting and valuable and contribute significantly to our understanding of antibiotics fate in the soil environment under the influence of different physiochemical factors. The presented paper is surely of interest for the potential readers of IJERPH.

Nevertheless there are several issues that need to be adressed before it can be accepted in the present form (please see the list below). Therefore I suggest the major revision f the paper.

Firstly, the manuscript needs the professional English proofreading as there are a multitude of mistakes that makes the paper difficult to follow at some stages. 

Please see just few from the Abstract part listed below:

In Abstract, Line 11: "Paddy soils were..." I believe they still are widely distributed-please rethink the tense You use; similarly "soaked with water" I would substitute with "saturated with water" of "flooded"

Line 17 3) Soil organic particles nor clays were the major components determined the adsorption of SDZ on paddy soils -->it is not clear what the Authors try to implicate with the sentence

L21  SDZ co-adsorbing--> co-adsorption; , it could conclude--it could be cocluded

Secondly, despite the fact that the research design is appropriate, the data presentation should be improved and discussion part adjusted to the improvements - presented more clearly.

Below there is a list of suggestions/questions that should be taken into account:

L48-49 were conducted in soil solutions?--> I believe they were conducted in soil suspensions? As a matter of fact it was adsorption on the solid state that was the aim ot the batch studies and soil slution parameters were the variables

L 52  sorption behavior of SDZ on various soil components--> what do Authors mean by BEHAVIOUR? Mechanisms?

In Table 1 I would change "Whole soil" to "crude soil (sample)" and please explain what do You mean by Organic particle? How is it correlated with organic carbon content? Why is it lower in the organic phase than in the crude soil? Please check the calculations

L66 Sigma-Aldrin -->Sigma -ALDRICH

L72 "over than analytical grade" - what is that type of purity? ready for analysis?

L123 What do You mean by:" The pellets of centrifuging"?

Figure 1 Please specify which are kinetic studies results (the 3 diagrams on the top of the figure- You can indicate them with a,b,c letters), as based on the x axes it is obvious, but some Readers may not be familiar with this type of studies;

Similarly the 3 diagrams on the bottom of figure 1... I assume these are pairs with the kinetic diagrams above? It takes a while to match them as there is no information in the Figure caption - please give the extra numbering or letters to indicate what is shown on the Figure!

Or...separate it into 3 different Figures for a better visibility!

Then, when You describe the results from Line 160 please always refer to Figure 1 and the diagram that the Reader need to follow for example Fig 1c. That is essential to compare the results.

Line 161 as solution pH changing --> precisize the "change" --> "as solution pH decreased!"

Line 168-169 The further fitting indicated that the increase of DOM accelerated the reaction process and increased the intensity of SDZ sorption on paddy soil --> the sentence is very "general" and des not introduce any further explanation for the observed results. Please rearrange that

Line 173-174 This might suggest that the variation of soil solution pH, IC
and DOM content didn’t change the sorption mechanism. --> or rather that there were more than one dominant mechanism responsible for the adsorption of SDZ

L 180 whole-->crude

Figure 2 --> the same improvements suggestions as for the figure 1

In Discussion part please evaluate how the hydrophobic distribution of ionic, polar compound is confirmed by attenuation of C=O, C-O and O-groups.

Round 2

Reviewer 2 Report

The manuscript has been evaluated and corrected. I believe that in the present form it can be published by the IJERPH journal.

Hence, I suggest to accept it in the present form.